# Raman-Based Techniques in Medical Applications for Diagnostic Tasks: A Review

**DOI:** 10.3390/ijms242115605

**Published:** 2023-10-26

**Authors:** Yulia Khristoforova, Lyudmila Bratchenko, Ivan Bratchenko

**Affiliations:** Department of Laser and Biotechnical Systems, Samara National Research University, 34 Moskovskoye Shosse, Samara 443086, Russia; khristoforovajulia@gmail.com (Y.K.);

**Keywords:** Raman spectroscopy, surface-enhanced Raman spectroscopy (SERS), SERS-based immunoassays, chemometrics, biomarkers, spectral markers, diagnostics

## Abstract

Raman spectroscopy is a widely developing approach for noninvasive analysis that can provide information on chemical composition and molecular structure. High chemical specificity calls for developing different medical diagnostic applications based on Raman spectroscopy. This review focuses on the Raman-based techniques used in medical diagnostics and provides an overview of such techniques, possible areas of their application, and current limitations. We have reviewed recent studies proposing conventional Raman spectroscopy and surface-enhanced Raman spectroscopy for rapid measuring of specific biomarkers of such diseases as cardiovascular disease, cancer, neurogenerative disease, and coronavirus disease (COVID-19). As a result, we have discovered several most promising Raman-based applications to identify affected persons by detecting some significant spectral features. We have analyzed these approaches in terms of their potentially diagnostic power and highlighted the remaining challenges and limitations preventing their translation into clinical settings.

## 1. Introduction

In the last few years, the use of Raman spectroscopy has been widely used for analyzing the chemical composition of a sample [1,2,3,4,5]. It is well known that a Raman spectrum of a sample is a superposition of the signals caused by the contribution of the active molecules of the components presented in the tested sample. Changes in the quantitative and qualitative composition of a sample lead to changes in the Raman peak intensities, shapes, and locations that are typical for the sample. In particular, the intensity of the Raman peak is proportional to the concentration of the molecule presented in the tested sample. Therefore, analysis of the encoded information-rich spectra of the sample allows for extracting biological information on its relative chemical composition. These advantages of the Raman spectroscopy cover a wide range of application tasks when changes in the tested sample composition are related to disease development [1,2,6].

Numerous studies demonstrate significant advances in this area, particularly in regard to the analysis of the Raman spectral signatures that tend to change in a pathological state in contrast to the normal state of the body [7,8]. For example, Raman spectroscopy has been used to differentiate cancerous breast [9,10,11,12], prostate [13,14,15], lung [16,17], skin [18,19], cervical [20,21], colorectal [22,23], and other abnormal tissues. In addition, malignant tissues can be distinguished from benign ones [18,24], or different types of cancer can be classified [25]. Similar to cancer studies, Raman spectroscopy is frequently used to detect infectious [26] and non-infectious [27] diseases. In recent years, there has been considerable progress in detecting disease biomarkers in different body fluids such as blood, saliva, urea, etc., commonly referred to as liquid biopsy [28,29].

Although these impressive results have proved the high clinical potential of Raman spectroscopy in the past few decades, there remain some challenges that prevent the translation of Raman-based technologies into clinical settings [30,31,32]. To use Raman-based approaches as a diagnostic tool, it is necessary to determine whether the recorded spectral data would be reliable for prognostic tasks. Traditionally, biomarkers are used in routine clinical diagnostics and monitoring of different diseases [33,34,35]. Raman-based technologies, particularly surface-enhanced Raman spectroscopy (SERS), which is frequently utilized as a sensing platform [36,37], can detect specific biomarkers. In this context, Raman spectroscopy can have greater potential in comparison to the laboratory procedures currently utilized in clinical settings. On the other hand, several studies [30,38] have reported that Raman spectral features analyzed with different mathematical methods (chemometrics, machine learning, artificial intelligence, etc.) are able to detect a disease by means of the meaningful spectral differences between the disease and control cases without identifying the exact biomarkers.

Therefore, this review aims at identifying the advantages and limitations of different Raman-based approaches as diagnostic tools. To this end, this review analyzes the papers published in the past five years on Raman analysis of disease biomarkers, as well as on the Raman spectral landscape for disease detection.

## 2. Raman Spectroscopy

Raman spectroscopy is an optical method based on the inelastic scattering of photons by molecular bond vibrations and rotations. The peak location of the isolated vibration in the Raman spectrum or the rotation of the chemical bond in the molecule is commonly known. The Raman spectra of tissues and biofluids are a complex mix of molecules, and the vibrational frequencies associated with different functional groups in different molecules often overlap. This leads to the fact that the actual band of the functional group in a molecule can be shifted and become different compared to the band of an isolated vibration due to the internal biological environment and bonds with adjacent molecules [39]. Nevertheless, the functional groups related to specific molecules often provide some relatively narrow and well-resolved bands in the Raman spectra.

The main informative vibrations and rotations in biological molecules are observed in the range of 500–1800 cm^–1^, which is called the unique spectral “fingerprint” of a molecule. Some characteristic Raman peaks in the “fingerprint” area usually arise from nucleic acids, protein, lipids/phospholipids, amino acids, carotenoids, etc. Most characteristic functional groups in molecules have several peaks, which increases the reliability of their determination. The high-wavenumber region at 2700–3500 cm^–1^ also contains spectral signatures associated mainly with CH in proteins and lipids [39].

Even under normal conditions, the quantitative composition of tissue and body fluid components varies widely. The identification of differences in the structural organization of body fluids or biotissues under a pathological condition or during a disease is an extremely difficult task. Moreover, the Raman spectrum of a disease sample is not accompanied by new additional peaks, but by changes in the intensity level, width, and shift of the Raman bands. Therefore, qualitative analysis of the new spectral changes can provide important clues for the detection of specific biomarkers and diagnosis that require special mathematical tools (Figure 1). To reveal and retrieve important information, machine learning and chemometrics methods have recently come into wide use in Raman data analysis [30].

The SERS approach is a powerful Raman-based technique that is based on the thousand-fold amplification of the scattering signal onto a metal surface. The scattering signal can be amplified by the frequency modulation of excitation laser radiation and the localized surface plasmon resonance of a metallic nanomaterial. SERS requires some specific plasmonic nanostructures, usually the classic SERS substrates of gold, silver, or copper. After the excitation of the localized surface plasmon resonance, the SERS-generated signal from the sample is recorded similarly to a conventional Raman signal. The SERS approach is able to detect analytes in low concentrations. SERS biosensors are used to detect various biological samples and diseases, including cancers [2,6,9,10,11,12,13,14,15,16,17,18,19,20,21,22,23], kidney disease [40,41], neurogenerative disease, etc. [42,43].

## 3. Clinical Applications

This section deals with the latest results of Raman spectroscopy studies in medical diagnostics. We have analyzed the papers that highlight the application of Raman-based techniques to clinically relevant tasks, particularly to diagnosing cardiovascular disease as the world’s main cause of death, cancer, neurodegenerative disease, and coronavirus disease (COVID-19). All these diseases require improved quality of preliminary diagnosis, especially at the early stages due to the limitations or shortcoming of the current clinical diagnostics. In total, this review includes 48 papers; however, Raman spectroscopy is successfully used for other diagnostic tasks [2,26,44,45,46,47,48,49] and not limited to the examples given in this review. Table 1 and Table 2 present a brief summary of the reviewed studies and their main findings in terms of the biomarkers and spectral markers that are important for diagnosing the respective diseases.

### 3.1. Direct Detection of Disease Biomarkers

A biomarker is a biological molecule found in body fluids or tissues that is a sign of a normal or abnormal process, or of a condition or disease [92]. There is a wide range of disease biomarkers, including antibodies, proteins, nucleic acids, and peptides. Proteins are one of the most important biomarkers and variations in their concentration in blood, serum, saliva, or tissues can indicate disease [92]. In comparison to conventional Raman spectroscopy, the SERS technique has promising potential for detecting specific disease biomarkers due to its high sensitivity and capability of quantitative analysis. This approach suggests the use of functional SERS nanostructures (usually those of noble metals or noble metals combined with other structures) serving as plasmon-resonance-enhanced substrates; the Raman-active molecules are adsorbed next to the nanostructures, which allows the plasmon-enhanced Raman signal to be recorded (Figure 2) [93,94]. Moreover, such approaches as SERS-based immunoassays make it possible to simultaneously detect multiple biomarkers [50]. Therefore, SERS-based immunoassays have been found able to detect different clinically relevant biomarkers using different nanostructures and nanoparticles as well as combinations of Raman-active molecules (Figure 2).

#### 3.1.1. Cancer

Detecting the low amounts of cancer biomarkers that can be present at the early stages of the disease can significantly raise the survival prospects of cancer patients. Cancer biomarkers include a broad range of biochemical entities, such as nucleic acids, proteins, sugars, lipids, and small metabolites, cytogenetic and cytokinetic parameters, as well as whole tumor-specific cells circulating in the body fluid [95]. Several papers [50,51,52,53,54,96] have reported on the development of SERS-based assays for detecting cancer antigens in human serum in low clinically relevant concentrations.

A point-of-care testing sensor based on SERS assay [50] was developed to detect multiplex prostate biomarkers such as prostate-specific antigen (PSA), carcinoembryonic antigen (CEA), and alphafetoprotein (AFP). Raman shift intensities at 593 of NBA, 1074 of 4-MB, and 1343 cm^−1^ of 4-NBT correlate with concentrations of PSA, CEA, and AFP, respectively. The proposed biosensor shows a wide linear dynamic range (LDR) with detection limits of 0.37, 0.43, and 0.26 pg mL^−1^ for PSA, CEA, and AFP. The immunoassay was tested on five different human serum samples.

A novel SERS-based multiplex immunoassay was proposed in 2018 by Wang et al. [51] to detect multiple tumor markers of prostate specific antigen (PSA) and α-fetoprotein (AFP) in human serum samples. They used the 1590 cm^−1^ peak of 4 MBA and the 1340 cm^−1^ peak of 4 NTP to quantitatively evaluate the concentration of PSA and AFP tumor markers, respectively. They found out that the SERS-based multiplex immunoassay had a wider dynamic linear range from 10 fg mL^−1^ to 400 ng mL^−1^ and the limits of detection for PSA and AFP were 3.38 fg mL^−1^ and 4.87 fg mL^−1^, respectively.

A SERS-lateral flow assay biosensor for analyzing the microRNAs (miRNAs) associated with lung cancer was developed by Cao et al. [50]. They looked into SERS peak intensities of 4-ATP at 1083 cm^−1^ and DTNB at 1330 cm^−1^ to quantify miR-196a-5p and miR-31-5p in clinical serum from 120 lung cancer patients and 30 healthy subjects. The detection limits of miR-196a-5p and miR-31-5p were as low as 1.171 nM and 2.251 nM in the phosphate buffer, and 1.681 nM and 2.603 nM in the human serum.

A SERS-based immunosensor was created by Panikar et al. [53] for detecting a soluble B7-H6 biomarker in blood serum from cervical cancer patients. They used ATP as a Raman probe molecule, which is stable at a wide range of pH, and analyzed the Raman intensities at 731 cm^−1^ correlated with B7-H6 concentrations. The biosensor was experimentally tested in nine cervical cancer patients and one healthy volunteer with a detection limit for the B7-H6 biomarker of 10^−14^ M or 10.8 fg mL^−1^.

Tian et al. [54] proposed novel kinds of visible-light-photoactive, SERS-sensitive, and magnetic-separable Fe_3_O_4_@TiO_2_@AuNW nanocomposites for the recyclable immunoassay of CA19–9 in the serum of colorectal cancer patients. The study obtained a low CA19–9 limit of 5.65 × 10^−4^ IU mL^−1^ and a wide linear range from 1000 to 0.001 IU mL^−1^ using the proposed (SERS-based) immunoassay. The ability of the SERS-based immunoassay to quantify CA19–9 was proved in eight clinical serum samples of colorectal cancer patients and the results are comparable to those obtained using the CLIA method.

#### 3.1.2. Cardiovascular Diseases (CVDs)

According to the World Health Organization [97], CVDs are the leading cause of death globally, taking an estimated 17.9 million lives each year. Cardiac biomarkers play a key role in the accurate diagnosis and management of CVDs, as well as in their prognosis. Different biomarkers have been proposed for diagnosing and monitoring CVDs. The most useful CVD indicators includes cardiac troponin I (cTnI), brain natriuretic peptide (BNP), its N-terminal pro-BNP precursor (NT-proBNP), the D-dimer, myoglobin, creatinine kinase myocardial band (CK-MB, an isoform of creatinine kinase), and C-reactive protein (CRP) [36,98,99].

The aptamer-immobilized Au nanoplate platform reported in [55] is capable of sensitive and selective sensing of cTnI. The SERS signals of Cy5 at 1580 cm^−1^ correlate with the concentration of cTnI. The aptamer-immobilized Au nanoplate platform was applied to the clinical diagnosis of AMI in nine clinical serum samples (three samples from healthy people and six samples from AMI patients). The proposed platform is able to detect cTnI at a concentration of 100 aM (2.4 fg/mL) in buffer solution and at a concentration of 100 fM (2.4 pg/mL) in serum. The aptamer-immobilized Au nanoplate made it possible to diagnose nine clinical samples more accurately than the conventional ELISA.

A new SERS-based sandwich immunoassay technique is proposed in [56] for ultrasensitive detection of AMI biomarkers using a new SERS nanoprobe and a gold patterned chip. The SERS signal intensity at 1615 cm^−1^ of malachite green isothiocyanate was analyzed as a function of the logarithm for the concentrations of cTnI and CK-MB. The detection limits of cTnI and CK-MB determined by the SERS assay were 8.9 pg mL^−1^ and 9.7 pg mL^−1^. The clinical applicability of the proposed SERS immunoassay was tested on five clinical serum samples from patients with acute myocardial infarction.

A lateral flow assay (LFA) based on SERS nanotags was presented in [57]. The SERS spectra were recorded from 50 human serum samples from patients with acute myocardial infarction (AMI). The SERS signal at 592 cm^−1^ was used for quantitative analysis of myoglobin, cardiac troponin I (cTnI), and creatine kinase-MB isoenzymes in various concentrations. Later, the same research team [58] reported on an improved novel SERS-based lateral flow assay (LFA) for rapid quantification of three biomarkers—myoglobin, cardiac troponin I (cTnI), and creatine kinase-MB—based on 448, 592, and 1510 cm^−1^ SERS bands. The designed optical sensor was tested in five human serum samples of patients with AMI and covered the linear dynamic range of 0.02−90, 0.01−50, and 0.01−500 ng/mL for creatine kinase-MB, cTnI, and myoglobin, respectively.

A SERS immunoassay for cardiac troponin was developed in [59] to diagnose acute myocardial infarction. The diagnostic potential of this sensor was proved in 50 serum samples from AMI patients. The concentration of cTnI biomarkers was quantified using the characteristic Raman peak intensities at 1075 cm^−1^ of the Raman-reporter molecules (4 MBA). The sensor achieved a detection limit of cTnI equaling 9.80 pg mL^−1^, which proved that the SERS immunoassay had good potential for AMI early diagnosis.

SERS-based immunosensors are also used to detect N-terminal pro-brain natriuretic peptide (NT-proBNP) as a specific biomarker of heart failure and other cardiac diseases [60,100]. For example, [60] provides an analysis of NT-proBNP for diagnosis cardiorenal syndrome with other biomarkers. The proposed SERS-based sandwich immunoassay is able to detect cTnI, NT-ProBNP, and neutrophil gelatinase-associated lipocalin (NGAL) simultaneously. The cTnI, NT-ProBNP, and NGAL were simultaneously detected using well-defined SERS signals of DTNB, NT, and 4-MBA at characteristic peaks of 1323 cm^−1^, 1363 cm^−1^, and 1584 cm^−1^, respectively. They achieved detection limits estimated to be 0.76, 0.53, and 0.41 fg mL^−1^ for cTnI, NT-ProBNP, and NGAL, respectively, which is one order of magnitude higher than that of the electrochemiluminescence detection method. The developed sensor was estimated on 10 clinical blood samples.

#### 3.1.3. Neurodegenerative Diseases

The growing number of patients with neurodegenerative disease has required the development of effective early diagnosis and monitoring techniques [101]. Raman spectroscopy has made it possible to research neurodegenerative disease by screening biological fluids, such as blood serum, saliva, tears, etc., and shown a significant potential for early diagnosis of Alzheimer’s disease (AD) and Parkinson’s disease (PD). The biomarkers of AD are essentially the occurrence of pathological aggregates of proteins, including the amyloid-β peptide that accumulates in the inter-neuronal space, forming amyloid plaques and tubulin-associated unit (tau) proteins that can transport into the vascular system.

Combining the SERS technique with magnetic separation has brought about new advances in detecting the biomarkers of AD. For example, a SERS-based immunoassay for amyloid-β detection was demonstrated by Back et al. [42]. Their technique involves the use of gold-coated magnetic nanoparticles immobilized by antibodies amyloid-β and SERS probes based on gold nanoparticles modified with detection antibodies that are labeled with 3,3,diethylthiatricarbocyanine iodide (DTTC) as SERS-sensitive platforms for amyloid-β recognition. The SERS peak at 1236 cm^−1^ was investigated to find the correlation between the Raman signal and amyloid-β concentration. The proposed detection strategy showed that the SERS-based immunoassay was able to detect amyloid-β with a much lower limit of 1 fM.

A chiral *d*-Pt@Au TNR SERS platform [43] can be applied to detect the Aβ monomers and fibrils that are the hallmarks of AD. Hollow chiral nanorings were synthesized to determine the process of Aβ42 protein misfolding and aggregation during neurodegenerative diseases and to quantify the secondary structure of these proteins. The 1245 and 1266 cm^−1^ band (in the amide III region) indicates the presence of the β-sheet and α-helical structure in the Aβ42 fibril and monomer. Therefore, the Raman intensity at 1245 cm^−1^ showed an excellent linearity with the concentration of Aβ42 fibrils while the Raman intensity at 1266 cm^−1^ was used to reflect the concentration of the Aβ42 monomer. The achieved detection limit was 0.045 × 10^−12^ m and 4 × 10^−15^ m for the Aβ42 monomer and fibrils, respectively.

Another SERS-based immunoassay was proposed [61] for sensitive and multiplexed detection of Aβ40 and Aβ42 AD biomarkers. The potential of this platform was demonstrated by using human serum with spiked Aβ40 and Aβ42 at concentrations ranging from 0.1 to 1000 pg mL^−1^. The SERS intensity of 4-FBT at 625 cm^−1^ increased along with the increase in Aβ40 concentration in the human serum, while the SERS signal of 4-BBT at 492 cm^−1^ increased with the increase in Aβ42 concentration, which made it possible to detect the amounts as low as 0.25 pg mL^−1^.

A robust and convenient SERS biosensing platform for simultaneous detection of Aβ(1−42) oligomers and tau protein was constructed [62] by using different Raman dye-coded polyA aptamer-AuNPs (PAapt-AuNPs) conjugates. The proposed strategy displayed excellent analytical performance in detecting tau protein and Aβ(1−42) oligomers with a detection limit of 4.2 × 10^−4^ pM and 3.7 × 10^−2^ nM, respectively.

A SERS-enabled lab-on-a-chip platform for simultaneous quantification of α-synuclein, phosphorylated tau protein 181, osteopontin, and osteocalcin, which are the Parkinson’s disease (PD) biomarkers, was fabricated [63]. The SERS signals of α-syn at 1508 cm^–1^, p-tau-181 at 1080 cm^–1^, OPN at 1508 cm^–1^, and OCN at 1080 cm^–1^ increased with an increase in their concentrations. The achieved limit of detection for α-syn, p-tau-181, OPN, and OCN was 0.82, 0.88, 0.91, and 0.97 pg/mL, respectively, when they were dissolved in serum in different concentrations. Also, the sensing capability of the designed platform was demonstrated in six samples of PD mice serum at different stages. The authors concluded that the proposed platform for simultaneous detection of PD biomarkers presented great potential as an alternative PD screening approach.

Zhang et al. [64] proposed a SERS-based sensing strategy for dopamine (DA) detection, abnormal variations in which can be linked to serious neurological disorders such as schizophrenia, Huntington’s disease, Alzheimer’s disease, and Parkinson’s disease. The SERS intensity of 3-mercaptophenylboronic acid (3-MPBA) as a Raman reporter at the 998 cm^–1^ band was analyzed to measure the DA concentration The study obtained a DA detection limit of 0.3 pM and a wide linear range of 1 pM to 1 μM. In addition, they applied the proposed SERS assay to the monitoring of DA in two CSF samples of patients, and the results suggested good accuracy and acceptable precision of the method in the detection of DA for physiological and pathological studies.

#### 3.1.4. SARS-CoV-2 Infection

The ongoing pandemic of coronavirus disease 2019 (COVID-19) caused by severe acute respiratory syndrome coronavirus 2 (SARS-CoV-2) is a heavy burden on the healthcare system worldwide. Globally, as of Aug 2023, there were over 768 million confirmed cases of COVID-19, of which 6.9 million cases were fatal, and the numbers keep growing [102]. Since the first reported case of COVID-19, there have been numerous publications aimed at developing diagnostic techniques to detect SARS-CoV-2.

IgM, IgG, and antigen are mentioned as the most reliable SARS-CoV-2 biomarkers [103,104]. The biochemical biomarkers include D-dimer, troponin, and creatine kinase [104]. Ref. [104] also reports on such inflammatory biomarkers as C-reactive protein (CRP), interleukin-6 (IL-6), procalcitonin (PCT), and ferritin (FT) reflecting the severity of COVID-19.

Several biosensors have been developed for the early detection of SARS-CoV-2 biomarkers [105], including those based on the SERS technique [106]. First, a SERS-based lateral flow assay for rapid and sensitive COVID-19 detection was demonstrated in [65]. As high-performance SERS tags, they used a SiO_2_@Ag nanocomposite labeled with dual layers of DTNB. The SERS-based assay was capable of simultaneous detection of anti-SARS-CoV-2 IgM and IgG based on the Raman intensity of 1328 cm^−1^. The assay was verified using 68 clinically tested clinical serum specimens that consisted of 19 positive and 49 negative samples of COVID-19 with 100% accuracy and specificity and a 1.0 pg/m LOD.

A SERS-based immunoassay involving an antibody pair, SERS-active hollow Au nanoparticles (NPs), and magnetic beads [66] was claimed effective for detecting SARS-CoV-2. The proposed assay was able to identify SARS-CoV-2 in the human nasopharyngeal aspirates of 15 patients by means of its correlation with the Raman intensity at 1170 cm^−1^. The detection limit for the SARS-CoV-2 antigen was 2.56 fg/mL.

Serebrennikova et al. [67] developed a lateral flow immunoassay (LFIA) of a specific antigen of the causative agent of COVID-19 by means of gold nanoparticles with immobilized antibodies and 4-mercaptobenzoic acid as the surface-enhanced Raman scattering (SERS) nanotag. The developed SERS-based LFIA was validated using spike receptor-binding domain (RBD) protein determination in inactivated SARS-CoV-2 virions with a low detection limit of 0.1 ng/mL and using a characteristic band of 1076 cm^−1^.

The SERS-based COVID-19 biosensor proposed in [68] is designed for detecting the SARS-CoV-2 virus using the immunoreaction between the SARS-CoV-2 spike antibody SERS-immune substrate, the spike antigen protein, and the Raman reporter-labeled immuno-Ag nanoparticles. The biosensor can reportedly detect the SARS-CoV-2 spike protein at an ultra-low concentration of 6.07 fg mL^−1^ in untreated saliva using the intensity of the SERS peak at 1077 cm^−1^.

A SERS-based immunoassay strip was prepared [69] for the multiplex detection of anti-SARS-CoV-2 antibodies, IgM and IgG, by using gap-enhanced Raman nanotags as ultra-sensitive SERS probes. To determine the IgM and IgG concentrations, they chose a characteristic peak of 4-NBT at 1320−1340 cm^−1^ that reflected a change in the structure and a reduction in the electron density of the nitro group influenced by the surface-conductive Au shell. The detection limits of IgM and IgG were found to be 1 ng/mL and 0.1 ng/mL, which is about 1/100 of commercially available immunoassay strips.

### 3.2. Spectral Landscape Analysis

In contrast to analyzing the exact biomarker contribution to the defined wavelength, it is possible to analyze the whole registered spectral region of tested samples and find some spectral differences between the disease and normal samples. In this case, all the metabolites contained in the tissues and biofluids more or less contribute to the analyzed spectra. Numerous studies report considerable advances in regard to the analysis of the changes in Raman spectral signatures due to disease in contrast to healthy cases or different stages/types of the disease [2,6,7,8,9,10,11,41]. The spectral differences between the disease and control cases can be attributed to changes in such Raman active molecules as proteins, lipids, phospholipids, nucleic acids, carotenoids, etc., that are present in various forms of biological samples. Due to the contribution of the biomolecule functional groups to different Raman peaks [3,4,5], the spectral differences are observed in different bands at a time. Advanced statistical analysis algorithms are typically applied to analyzing the spectral datasets presented in the matrix of Raman intensities at the corresponding discrete wavenumbers, which makes it possible to detect significant spectral differences and similarities between different classes [30]. Therefore, to define chemically relevant information and to predict disease, the significance of different spectral signatures is determined. For example, the variable importance in projection (VIP) distribution is estimated to highlight the informative spectral predictors of tumors in the statistical model that are more important for classifying different disease states (Figure 3). A higher relative intensity in VIP score indicates that the predicted variable is more significant. In this connection, of particular interest is to establish whether the emerging biochemical changes measured using Raman spectroscopy are specific to a certain type and location of a disease.

#### 3.2.1. Cancer

Several studies employed another approach to detecting cancer patients by means of Raman spectroscopy. Instead of trying to capture the clinical biomarkers in biomaterials, they analyzed significant spectral differences (spectral markers) between the cancer and control cases.

Raman spectroscopy was able to find biomarkers in different biological sources, most commonly serum blood, plasma blood, whole blood, and, less commonly, saliva, urine, etc. for breast cancer diagnostics. Nargis et al. [9] reported on employing surface-enhanced Raman spectroscopy and Raman spectroscopy to diagnose breast cancer using clinically diagnosed serum samples from 17 breast cancer patients and 12 healthy individuals. Significant differences between the cancerous and healthy serum samples were found for the following Raman spectral features: 737 cm^−1^ (DNA, tryptophan, d (ring)), 766 cm^−1^ (ring breathing mode of cytosine and thymine), 784 cm^−1^ (cytosine), 950 cm^−1^ (single bond stretching for amino acids), 1191 cm^−1^ (L-valine), 1313 cm^−1^ (CH_3_CH_2_ twisting mode of lipid), 1424 cm^−1^ (deoxyribose (B, Z-marker)), 1542 cm^−1^ (amide II), and 1628 cm^−1^ (L-serine) bands. Such biomolecules as DNA bases, proteins, and lipids proved to be direct breast cancer markers dominantly present in the serum samples of breast cancer patients in contrast to healthy blood serum samples. Kopec et al. [70] reported on the application of Raman spectroscopy and Raman imaging to distinguish between glycosylated and non-glycosylated proteins in normal and cancer breast tissues involving seven patients with breast cancer. A comparison of the spectra of the normal and cancerous tissues revealed that: (1) in contrast to normal breast tissue, the cancerous tissue does not express the presence of glycan-rich regions; and (2) protein-rich regions dominate the structure of the tumor mass in contrast to the normal tissue that is abundant in adipose tissue. Thus, Raman profiles were able to detect the markedly deregulated metabolism of proteins, lipids, and glycans in breast (adenomacarcinoma) cancer. In Ref. [71], an increase in the DNA/RNA signal intensity within the fingerprint region (600–1800 cm^−1^) and the global loss of a high wavenumber signal (2800–3200 cm^−1^) are marked as warning signs of breast tumors.

Zheng et al. [72] reported that Raman spectroscopy revealed remarkable biochemical alterations in cancer lung tissues in comparison with normal tissues. They analyzed 16 normal and 50 cancer lung tissues, including 40 adenocarcinomas and 10 squamous carcinomas. Their results showed that the most prominent biochemical alterations between the cancer and control cases were mostly observed in the DNA/RNA and protein regions, while the most prominent biochemical differences between the adenocarcinomas and squamous carcinomas mostly appeared in the protein regions. As significant differences, they reported increased levels of saturated and unsaturated lipids and decreased ratios of both protein to lipid and nucleic acid to lipid in cancer in contrast to normal lung tissues. Moreover, the protein-to-nucleic-acid ratio was significantly greater in cancer tissues than in normal tissues. In Ref. [17], Zhang et al. applied the SERS technique to predict lung diseases by means of serum samples. The practicability of the proposed method was proved by detecting serum samples from 50 lung cancer patients and 50 normal healthy individuals. The results demonstrated the following significant differences in lung cancer diagnosis by means of Raman measurements: (1) for the lung cancer patients, the SERS signals of tyrosine (638 cm^−1^), L-serine (813 cm^−1^), and L-tryptophan (1207 cm^−1^) in the serum showed lower intensity with decreased content of amino acid; (2) the DNA/RNA level (725 cm^−1^, 1573 cm^−1^) in the lung cancer serum was higher than that in normal serum; (3) the collagen and phospholipid 1445 cm^−1^ band and the phenylalanine 1580 cm^−1^ band were lower in the normal serum than in the cancerous serum. In total, the intensity of most Raman peaks in the serum of the lung cancer patients was reported to increase, which might be due to the abnormal proliferation of cancer cells [17].

In Refs. [72,73], Chen et al. demonstrated that Raman spectroscopy combined with a deep learning model can quickly and accurately diagnose lung cancer by means of the serum analysis of 36 lung cancer patients and 34 control subjects. The results showed that proline and valine (950 cm^−1^) can be potential biomarkers of lung cancer because they are related to the production and metabolism of lung cancer tumor cells [107]. The change in phenylalanine level (1006 cm^−1^) can be attributed to the growth and metastasis of a variety of malignancies, including lung cancer. Furthermore, it was reported that proline metabolites are involved in the formation of lung cancer. A substantially higher Raman intensity (1343 and 1448 cm^−1^) in the lung cancer patients in contrast to the control subjects can be attributed to changes in glucose and lipid/protein levels, respectively, due to the growth and metabolism of tumor cells. The serum levels of β-carotene, which has anticancer effects, in patients with lung cancer are substantially lower than those in control subjects for 1155 and 1517 cm^−1^. Moreover, the impact of the serum cholesterol level on lipid metabolism is 1669 cm^−1^ higher in patients with lung cancer. In Ref. [74], Zhang et al. reported that the differences in the SERS tissue for glucose (913 cm^−1^), DNA (1079 and 1421 cm^−1^), carotenoids (1152 and 1514 cm^−1^), lipids (1089, 1453, and 1652 cm^−1^), and proteins (1152, 1453, and 1585 cm^−1^) are the most striking aspects when comparing cancerous and normal lung tissues. Specifically, the glucose, DNA, lipid, and protein contents are higher in the cancerous tissues than in the normal ones. Moreover, cancerous lung tissues contain a markedly higher concentration of carotenoids (1152 and 1514 cm^−1^) due to their higher C=C stretching vibration intensities than those of the control tissue. Interestingly, [72,73] reported that β-carotene had lower intensities at the same peaks of 1155 and 1517 cm^−1^ in the blood serum samples of lung cancer patients, which might be due to the differences in carotenoid metabolism between different tissues.

Several studies have demonstrated that Raman spectroscopy is potentially suitable to detect the malignization of lung tissues and to understand cancer progression using different biological form samples. In Refs. [25,74], an increase in the Raman bands of proteins and lipids was primarily observed in both liquid and tissue samples. Some studies reported the importance of vibrations in such molecules as carbohydrates, β-carotene [73,73], tyrosin [17], L-serine [17], etc. However, other studies present the opposite results. For example, Ref. [17] demonstrates higher Raman intensities in lung serum samples, while [108] reports on weaker spectral lines in lung cancer patients in comparison to healthy individuals.

In 2020, Medipally et al. [28] applied Raman spectroscopy to prostate cancer diagnosis for 43 prostate cancer patients and 33 healthy control volunteers. They claimed that Raman spectral analysis of blood plasma and lymphocytes exhibited consistent changes related to proteins, lipids, and nucleic acids in plasma from healthy donors and prostate cancer patients with varying Gleason scores. Aubertin et al. [75] employed Raman spectroscopy for diagnosing and possibly grading prostate cancer by analyzing 32 fresh and non-processed post-prostatectomy prostatic specimens. They found significant differences between the prostatic and extraprostatic samples at the peaks related to lipids. Moreover, the prostatic tissue samples showed higher amplitudes for the peaks associated with proteins and nucleic acids.

Xia [76] used SERS analysis to identify malignant and benign thyroid nodules by using blood serum samples from healthy volunteers (n = 22), patients with benign nodules (n = 19), and patients with malignant nodules (n = 22). One of the main findings was that the Raman peaks related both to amino acid metabolism and DNA/RNA metabolism demonstrated great potential for identifying thyroid nodules, while the DNA/RNA metabolism Raman peaks can further be used to assess the progression of thyroid nodules. More specifically, the SERS bands of proline, valine (957 cm^−1^), phenylalanine (1574 cm^−1^), and amide I (1654 cm^−1^) were higher in the thyroid nodular group, suggesting an increase in amino acids in the blood serum from patients with thyroid nodules. Furthermore, differences in the spectral signatures associated with nucleic acid (1332 cm^−1^) and adenine (725 cm^−1^) are observed due to abnormal DNA/RNA metabolism in the patients with thyroid nodules. Sbroscia et al. [77] enrolled 30 subjects with thyroid nodular pathology to examine the diagnostic efficiency of Raman spectroscopy. They noticed a distinctive set of peaks that characterize different type of tissues. The healthy tissue spectra were characterized by prominent fingerprints of cytochrome *c* at 1600, 747, 1120–1128, and 1301 cm^−1^. The spectra of the classical papillary carcinoma showed the presence of carotenoids bands at 1003, 1155, and 1516 cm^−1^, which are absent in healthy tissues. The follicular carcinoma spectra demonstrated enhanced cytochrome *c* bands, whereas the spectrum of the papillary follicular carcinoma showed fingerprints of carotenoids along with enhanced cytochrome *c* bands. In Ref. [78], cytochrome is also mentioned as a possible biomarker of thyroid biomaterial to recognize pathology. Cancerous follicular thyroid carcinoma cells are reported to contain increased populations of lipid-containing components and decreased populations of cytochrome-containing components relative to regular human thyroid follicular cells. In Ref. [79], Raman spectroscopy was used to measure the serum samples taken from 34 thyroid dysfunction patients and 40 healthy volunteers. As spectral markers of thyroid dysfunction, the authors suggest the changes in the ring vibration of glycerol at 630 cm^−1^, which was higher in abnormal thyroid function patients, maybe due to a decrease in lipid metabolism and the C-C symmetric stretch of phenylalanine at 1004 cm^−1^, which was lower in the normal thyroid function subjects. The Raman bands at 1154 cm^−1^ and 1513 cm^−1^ were assigned to the vibrations of carotenoids and were lower in the thyroid dysfunction patients than in the healthy subjects.

#### 3.2.2. Cardiovascular Diseases (CVDs)

The reviewed studies have demonstrated that CVD diagnosis not only involves the development of optical biosensors and assays to detect specific biomarkers using Raman signals but also the analysis of the changes in biomaterial spectral profiles during disease progression. To diagnose CVDs, the Raman spectral features are recorded for different biomaterials, including those of the heart, blood, and other body fluids. For example, [80] applied Raman spectroscopy to diagnose coronary heart disease (CHD) with a most unexpected biological material—urine. When the recorded urine SERS spectra from 87 patients with CHD and 20 healthy humans were compared with the clinical data, it was found that the Raman peak at 1509 cm^−1^ attributed to platelet-derived growth factor-BB proved to be efficient for distinguishing between the two analyzed groups.

Raman spectroscopy can also be employed as a diagnostic tool to detect acute myocardial infarction (MI). In Ref. [81], whole blood and blood serum were analyzed as a sample to examine inflammation in 10 acute MI patients in comparison to 10 healthy volunteers. The study revealed that the Raman peaks that correspond to phenylalanine (1000 cm^−1^) and tyrosine (825 cm^−1^) can be exploited as useful spectral signatures to probe the inflammation caused by MI. Li et al. [8] used SERS to measure human urine to diagnose coronary heart disease (CHD). They examined the urine samples of 157 CHD patients and 63 healthy controls and found differences in intensity at nine Raman peaks (1223/1243/1272/1463/1481/1516/1536/1541/1550 cm^−1^) between CHD and healthy controls in their average SERS spectrum.

The viability of ischemic myocardial tissue can be evaluated using Raman spectroscopy in patients undergoing cardiac surgery. Yamamoto et al. [82] applied Raman spectroscopy to the human heart to detect myocardial viability during surgery, involving five patients. They identified the key signatures of the Raman spectra for evaluating myocardial viability, which represent the contribution of heme proteins (755 cm^−1^) and collagen (the peak position was about 2930 to 2950 cm^−1^).

#### 3.2.3. Neurodegenerative Diseases

Recent studies have revealed that Raman spectroscopy was successfully applied to detect biochemical changes during neurodegenerative disorders. It is interesting to note that useful chemical information on neurodegenerative diseases can be obtained using a Raman-based technique applied to different biofluids such as tears, blood, and saliva. Cennamo [83] proposed a method based on SERS of tears for diagnosing neurodegenerative pathologies, including different forms of dementia and AD. The study involved 18 AD patients, 8 patients with mild cognitive impairment, and 6 control volunteers. The classes related to lactoferrin and lysozyme protein components manifested some spectral differences. Furthermore, identifying the I_1342_/I_1243_ ratio assigned to the C-H deformation and amide III β-sheet provided some quantitative information on the changes related to pathological conditions.

For diagnosing dementia of the Alzheimer’s type, Carota et al. [84] employed Raman spectroscopy to analyze a blood serum sample from 26 healthy controls and 31 patients and found that carotenoid levels of 1000, 1154, and 1519 cm^−1^ can be informative in terms Alzheimer disease of progression since the intensity of the carotenoid signals decreased from the early stage to the more severe stages.

Ryzhikova et al. [85] proposed a novel method to detect AD by analyzing blood serum using SERS in combination with multivariate statistical analysis. Serum samples were collected from 48 individuals, including 10 patients with Alzheimer’s disease at either mild or moderate stages, 5 patients with Lewy body dementia, 10 patients with Parkinson’s disease dementia, 3 with frontotemporal dementia, and 10 healthy controls. The spectroscopic markers that were selected using a genetic algorithm (GA) as the most useful for the correct classification of the measured SERS spectra, capable of distinguishing between AD and healthy controls through the detection of specific changes in particular nucleic acids, saccharides, and protein content in the blood serum, were specifically the 674–710, 899–973, 1087–1123, 1162–1199, and 1613–1649 cm^−1^ regions. Meanwhile, the 448–485, 523–560, 598–635, 759–785, 861–936, 1124–1161, 1312–1386, 1613–1649, and 1688–1724 cm^−1^ regions were selected for distinguishing between Alzheimer’s disease and other forms of neurodegenerative dementia.

Carlomagno et al. [86] estimated the potential of Raman analysis for diagnosing Parkinson’s disease. They applied machine and deep learning analysis of the Raman signals to the saliva of 23 Parkinson’s disease (PD) patients, 10 AD patients, and 33 healthy controls. They identified some evident differences for the PD compared to the healthy and AD subjects, principally regarding the peaks and bands related to proteins (829, 939, and 1001 cm^–1^ are signals from specific amino acids, while 1102 and 1346 cm^−1^ are due to the amide bands), nucleic acids (1244 cm^–1^), glycoproteins/saccharides (850 and 1444 cm^−1^), and lipids. The results suggest that the high concentrations of α-synuclein, heme-oxygenase-1, protein damaged by ROS, and the cysteine protease DJ-1 found in the saliva of PD patients could explain the differences in protein. Changes in saccharides can be attributed to the altered metabolism of glucose and to carbohydrates in PD patients [86].

#### 3.2.4. SARS-CoV-2 Infection

For SARS-CoV-2 detection, many highly sensitive SERS-based sensors were proposed [65,66,67,68,69,105,106]; however, due to their complex design, the need for new simpler approaches to label-free analysis still persists. In this section, we present the diagnostic modalities of SARS-CoV-2 infection based on the analysis of Raman spectral data using machine learning and deep learning algorithms. The Raman data obtained from the label-free analysis of human biofluids proved to be effectively managed for classifying and predicting SARS-CoV-2 disease.

Karunakaran et al. [87] demonstrated a screening modality based on label-free SERS for scrutinizing the SARS-CoV-2-mediated molecular-level changes in saliva samples among 14 healthy and 14 COVID-19-infected subjects. The Raman peaks at 450 (thiocyanate), 1002 (Phenylalanine), 1224 (β sheet structure in amide III), 1453 (lipid), 1586 (phenylalanine), and 2126 cm^−1^ (thiocyanate) displayed variations between positive patients and healthy subjects.

Goulart et al. [88] proposed a technique for diagnosing COVID-19 by means of Raman spectroscopy involving blood serum samples from 10 COVID-19 positive patients and 10 COVID-19 negative patients. The principal component analysis variables showed spectral differences related to some biochemical alterations due to COVID-19, such as an increase in lipids, nitrogen compounds (urea and amines/amides), and nucleic acids, as well as a decrease in proteins and amino acids (tryptophan) in the COVID-19 group.

Carlomagno et al. [89] reported using SERS for measuring saliva samples from 30 current SARS-CoV-2-infected patients (COV^+^), 38 previously SARS-CoV-2-infected (COV^−^) patients, and 33 control subjects. They discussed that important differences between the control experimental group and the COV^+^  and COV^−^ groups were observed at 1048 and 1126 cm^−1^. The strong signals at the 1048 and 1126 cm^−1^ locations were explained by an environment rich in aromatic amino acids, especially tryptophan and phenylalanine. The same two peaks (1048 cm^−1^ and 1126 cm^−1^) have also been discovered as distinctive signals from coronaviruses, indicating their involvement in the viral protein structure or interactions with physiologically produced molecules.

Raman spectroscopy proved to be able to identify COVID-19 in 54 control serum samples and 40 COVID-19 samples with the presence of SARS-Cov-2 [90]. PCA analysis demonstrated that the main spectral differences between the positive IgM and IgG versus the control were as follows: (1) the features assigned to proteins including albumin were lower in the COVID-19 group as well as in the IgM/IgG and IgG positive group; (2) the features assigned to lipids, phospholipids, and carotenoids were higher in the COVID-19 group and in the IgM/IgG positive group; and (3) the features related to nucleic acids, tryptophan, and immunoglobulins were higher in the COVID-19 group.

In Ref. [91], Raman spectroscopy, in combination with multivariate and machine learning methods, was employed to detect the different SARS-CoV-2 antibody levels in patients infected with COVID-19. The study involved 47 patients infected with COVID. All the patients had their blood samples taken for antibody measurement 1 month after the first diagnosis, a second time after 3 months, and again after 6 months. The main finding was the possibility of using the Raman ranges between 1317 cm^−1^ and 1432 cm^−1^ and between 2840 cm^−1^ and 2956 cm^−1^ corresponding to proteins and lipids as markers that can examine the rate of recovery from the COVID-19 disease. Machine learning techniques demonstrated their capability to near-perfectly discriminate between patients one, three, and six months after COVID-19.

## 4. Critical Evaluation and Future Perspectives

Over the past few years, there have been numerous studies into Raman spectroscopy being applied to detecting various diseases. There is no doubt that Raman spectroscopy has proven its great potential for biomedical diagnostic tasks. Nevertheless, it is important to note that there are several items that need serious consideration in terms of its trustworthiness and diagnostic abilities.

In SERS-based sensors, isolated molecules and dyes are typically used as Raman reporters that provide intrinsic fingerprint molecular information [37]. As a rule, a characteristic signal of a Raman reporter at one Raman band is analyzed to indicate a target biomarker. The level of the target biomarker is analyzed using calibration curves and usually does not require special mathematical tools. In contrast, the methodology based on the recognition of the meaningful Raman spectral features and analysis of the whole spectral landscape requires special mathematical methods (statistical analysis, machine learning, or deep learning methods) to identify the most useful set of spectral data and to separate it from insignificant data and noise [109]. Unfortunately, our previous review of similar papers in this field [110] has revealed no generalized aspects when chemometrics methods are applied to the analysis of spectral data and the construction of statistical models. The high variability and low reproducibility of the spectral features in different studies for the same clinical task (Table 2) raise questions regarding the correctness of their definition. Moreover, many studies demonstrated a combination of Raman data and chemometrics [110,111,112,113] that might contain unsupported or invalid results due to typical errors related to their application. If we want to obtain reliable and reproducible results by using recognition and diagnostic technologies based on the methods of statistic modeling, our research projects and studies should follow certain requirements for statistical analysis application, particularly the requirements to avoid overfitted models and testing models built on unseen independent data, as well as the requirement to increase the experimental sample size to ensure significant and reliable results.

Nevertheless, the approaches based on spectral feature comparison for opposite classes (Table 2) are not able to distinguish specific biomarkers of a disease. The identified sets of important Raman differences generally reflect some small modifications in the tissue or body fluid composition that can be attributed to the changed levels of specific pathological and disease-stimulated substances. Such molecular components as proteins, lipids or phospholipids, amino acids, nuclear acids, and carotenoids marked as potential disease features constitute some broad functional groups of chemicals. They can only provide some nonspecific characterization of the changes in a biochemical substance that take place during disease, and the estimation of these changes is normally overgeneralized. One of the possible ways to identify some definite disease-specific markers would be a combination of Raman spectroscopy and certain accurate methods of biochemical analysis in clinical laboratories to compare the results of spectral analysis and quantitative chemical composition profiles. To obtain reliable and reproducible results, it is necessary to apply appropriate mathematical techniques that can make it possible to build accurate regression models between the Raman spectral features and an exact quantity of chemicals for a better understanding of what is occurring at the biochemical level of the molecules during disease progression.

Disease biomarkers can be detected and quantified by using sensors for SERS analysis. The proposed sensors require specific preparation of SERS platforms; nevertheless, the obtained results have demonstrated high accuracy and the ability to detect the target biomarkers in low concentrations. The reviewed studies on clinical biomarker identification summarized in Table 1 demonstrate large variations in different substrates, Raman reporters, and SERS band intensities correlated with the quantity of the target biomarkers. This variability also highlights the need to develop a standardized protocol on how to use the proposed SERS platforms. The main shortage here is the lack of reliable approbation of the developed sensors on clinical samples. In several studies [43,50,53,54,55,56,58,60,62,66,96], the achieved diagnostic abilities were estimated using buffer solutions with different concentrations of diluted target analytes, while their application to clinical samples with actual concentrations was tested on small sample sizes, which is not sufficient for validating their possible clinical use. It should be noted, however, that real field samples are more complex mixtures with different molecules that can lead to degraded detectability of the target biomarker. Therefore, clinical testing of SERS-based sensors is a crucial issue and should be enhanced for potential diagnostic tools.

This analysis does not provide an unambiguous and precise way to determine the most convenient or reliable approach to diagnostics and disease prognostics. At first, SERS platforms with nanotags seem to be appropriate and relevant for diagnostic purposes. However, over the past 10 years, a large number of optical SERS sensors proposed for quantifying biomarkers turned out to lack wide approbation on clinical samples, which is the reason for their serious complications and limitations [37,94]. On the other hand, a combination of Raman-based techniques and chemical analysis methods can possibly lead to finding some novel reliable biomarker panels in different biological forms, subject to the proper use of chemometrics methods for analyzing Raman spectral datasets.

To sum up, clinical application for diagnostic purposes requires finding a balance between the simplicity of implementation that nanoparticle-free Raman techniques have and the high accuracy of SERS platforms, which are able to detect low concentrations of the target analytes. It is equally important to use standardized protocols for uniform Raman applications and to develop special mathematical tools, particularly chemometrics, to retrieve and reveal clinically important information, which would ensure the trustworthy diagnostic effectiveness of Raman spectroscopy so that the task’s success should not depend on human subjectivity.

## 5. Conclusions

The growing number of studies in the field proves the common trend of using Raman-based technologies in biomedicine. This review examines the use of Raman-based approaches that can be suitable for clinically relevant diagnostic tasks. We have detailed the use of highly-sensitive SERS-based sensors that can detect and quantify various disease biomarkers. First, the developed SERS-based platforms have a promising analytical ability with a low detection limit for disease biomarkers and a wide dynamic detection range. Second, Raman applications based on spectral differences between disease and control cases are used for identifying spectral markers as a label-free method. Achievements in Raman spectroscopy can be a potential solution for creating a universal method for human tissue and biofluid analysis with rapid automated diagnostic feedback at the time of measurement. Finally, both approaches still have some limitations and complications that, despite their valuable achievements, still prevent Raman-based technologies from reaching the next level of reliable diagnostics and being translated into a clinical setting.

## Figures and Tables

**Figure 1 ijms-24-15605-f001:**
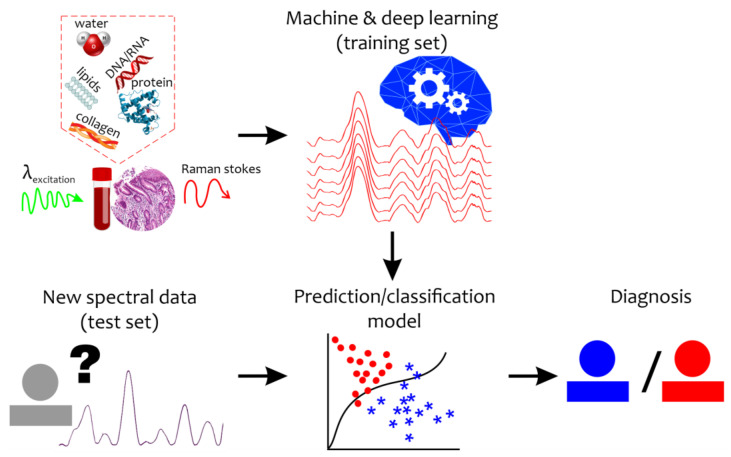
Schematic representation of the combination of the Raman-based technique and machine and deep learning algorithms for disease detection. “?”—new unseen spectral data with unknown diagnosis. Red circles mean abnormal states while blue asterisks mean healthy (control) states.

**Figure 2 ijms-24-15605-f002:**
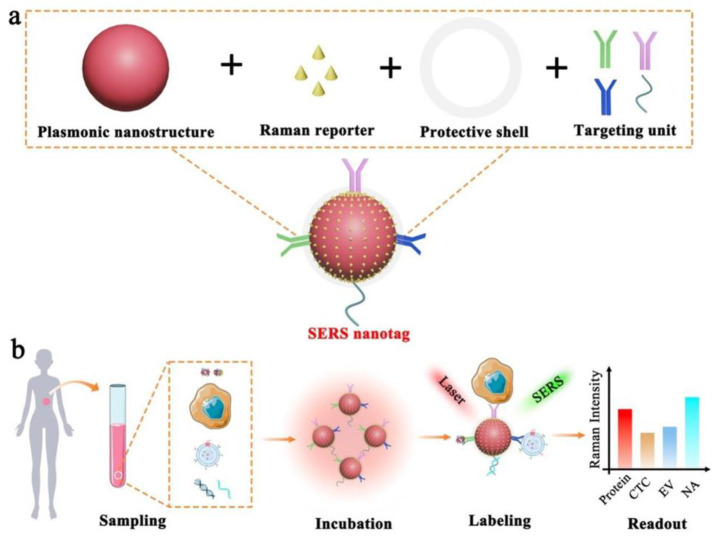
Schematic illustration of the SERS nanotag capable of highly sensitive and multiplexed detection of cancer circulating biomarkers. (**a**) The design of a typical SERS nanotag by means of four key components. (**b**) Application of the SERS nanotags to detecting protein, CTC, EV, and NA (reprinted under the CC BY 4.0 license from Ref. [37]).

**Figure 3 ijms-24-15605-f003:**
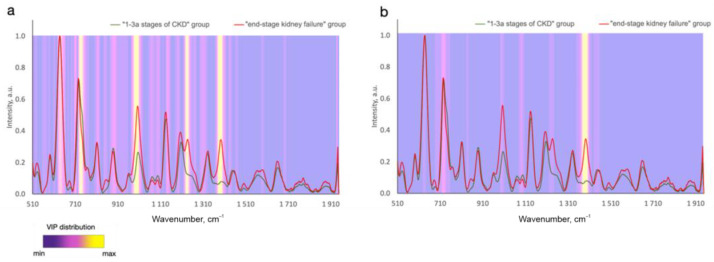
Variable importance in projection (VIP) distributions for (**a**) PLS-DA and (**b**) CNN statistical models that differ in serum SERS spectra between subjects at 1–3a stages of and end-stage chronic kidney disease (Reprinted with permission from [41] © The Optical Society]).

**Table 1 ijms-24-15605-t001:** Detection of disease biomarkers using Raman-based techniques.

Ref.	Disease	Biomarkers	Analyzed Wavelength (cm^−1^) and (Raman Reporters)	Clinical Samples
[50]	Prostate cancer	PSA	593 (NBA)	5 human serum samples
		CEA	1074 (4-MB)	
		α-fetoprotein	1343 (4-NBT)	
[51]	Prostate cancer	PSA	1590 (4-MBA)	4 serum samples
		α-fetoprotein	1340 (4-NTP)	
[52]	Lung cancer	miR-196a-5p	1083 (4-ATP)	30 healthy subjects and 120 lung cancer patients
		miR-31-5p	1330 (DTNB)	
[53]	Cervical cancer	B7-H6	731 (ATP)	10 serum samples
[54]	Colorectal cancer	CA19–9	1078 (4-MBA)	8 serum samples
[55]	AMI	cTnI	1580 (Cy5)	9 samples
[56]	AMI	cTnI	1615 (MGITC)	5 clinical serum samples
		CK-MB	1615 (MGITC)	
[57]	AMI	Myoglobin	592 (NBA)	50 serum samples
		cTnI	592 (NBA)	
		CK-MB	592 (NBA)	
[58]	AMI	Myoglobin	448 (Methylene blue)	5 human serum samples
		cTnI	592(NBA)	
		CK-MB	1510 (Rhodamine 6 G)	
[59]	AMI	cTnI	1075 (4-MBA)	50 serum samples
[60]	Cardiorenal syndrome	cTnI	1323 (DTNB)	10 clinical samples
		NT-ProBNP	1363 (NT)	
		NGAL	1584 (4-MBA)	
[61]	AD	Aβ-40	625 (4-FBT)	
		Aβ-42	492 (4-BBT)	
[42]	AD	Aβ	1236 (DTTC)	
[62]	AD	Aβ	1073 (AATP)	4 artificial cerebrospinal fluid samples
		Tau protein	1327 (DTNP)	
[43]	AD	Aβ42 fibrils	1245 (*d*-Pt@Au TNRs)	6 samples of cerebrospinal fluid
		Aβ42 monomer	1266 (*d*-Pt@Au TNRs)	
[63]	PD	p-tau-181, OCN	1080 (4-MBA)	6 samples of PD mice serum
		α-syn, OPN	1508 (R6G)	
[64]	ND	Dopamine	998 (3-MPBA)	2 human cerebrospinal fluid samples
[65]	SARS-CoV-2	IgM	1328 (DTNB)	19 COVID-19 patients and 49 healthy people
		IgG	1328 (DTNB)	
[66]	SARS-CoV-2	SARS-CoV-2 antigen	1170 (MGITC)	15 samples of human nasal fluid
[67]	SARS-CoV-2	SARS-CoV-2 spike receptor-binding domain	1076 (4-MBA)	
[68]	SARS-CoV-2	SARS-CoV-2 spike protein	1077 (4-MBA)	
[69]	SARS-CoV-2	IgM and IgG	1320–1340 (4-NBT)	IgM and IgG mixtures were added to bovine serum

Note: PSA = prostate-specific antigen; CEA = carcinoembryonic antigen; cTnI = cardiac troponin I; CK-MB = creatine kinase-MB; NT-ProBNP = N-terminal pro-brain natriuretic peptide; Aβ = amyloid beta; NBA = Nile blue A; 4-NBT = 4-nitrobenzenethiol; 4-MB = 4-Mercaptobenzonitrile; 4-MBA = 4-mercaptobenzoic acid; 4-NTP = 4-nitrothiophenol; 4-ATP = 4-aminothiophenol; 3-MPBA = 3-mercaptophenylboronic acid, DTNB = 5.50-dithiobis-(2-nitrobenzoic acid); MGITC = malachite green isothiocyanate; AMI = acute myocardial infarction; AD = Alzheimer’s disease; PD = Parkinson’s disease; ND = neurological disorder.

**Table 2 ijms-24-15605-t002:** Important spectral markers for detecting diseases using Raman-based techniques.

Ref.	Disease	Spectral Markers	Clinical Samples	Techniques
[9]	Breast cancer	DNA, proteins, and lipids can be considered direct breast cancer markers	Serum samples from 17 breast cancer patients and 12 healthy individuals	PCA, PLS-DA
[70]	Breast cancer	Metabolism of proteins, lipids, and glycans	Tissues of seven patients with breast cancer	Basic analysis methods and K-means cluster analysis
[71]	Breast cancer	DNA/RNA in the 600–1800 cm^−1^ and global loss of high-wavenumber signal (2800–3200 cm^−1^)	Breast tissues of five patients	Unsupervised K-means and stochastic nonlinear neural networks
[25]	Lung cancer	DNA/RNA and protein regions, saturated and unsaturated lipids, ratios of both protein to lipid and nucleic acid to lipid	16 normal tissues, 40 adenocarcinomas, and 10 squamous carcinomas	K-nearest neighbor and support vector machine
[17]	Lung cancer	Tyrosine (638 cm^−1^), L-serine (813 cm^−1^), and L-tryptophan (1207 cm^−1^), DNA/RNA level (725 cm^−1^, 1573 cm^−1^), collagen and phospholipids (1445 cm^−1^), phenylalanine (1580 cm^−1^)	Serum samples from 50 lung cancer patients and 50 normal healthy people	PCA-LDA
[72,73]	Lung cancer	Proline and valine (950 cm^−1^), glucose (1343 cm^−1^), lipids/proteins (1448 cm^−1^), β-carotene (1155 to 1517 cm^−1^), cholesterol level (1669 cm^−1^)	Serum samples of 36 lung cancer patients and 34 control subjects	Multilayer perceptron, recursive neural network, CNN, and AlexNet
[74]	Lung cancer	Glucose (913 cm^−1^), DNA (1079 and 1421 cm^−1^), carotenoids (1152 and 1514 cm^−1^), lipids (1089, 1453, and 1652 cm^−1^), and proteins (1152, 1453, and 1585 cm^−1^)	23 normal and 23 cancerous lung tissues	PCA-LDA
[28]	Prostate cancer	Proteins, lipids, and nucleic acids	Blood plasma and lymphocytes of 43 prostate cancer patients and 33 healthy control volunteers	PCA, PLS-DA, CLS
[75]	Prostate cancer	Proteins, lipids, and nucleic acids	32 fresh and non-processed post-prostatectomy prostatic specimens	Neural network, Kolmogorov–Smirnov test, F-test, *t*-test
[76]	Thyroid nodules	Both amino acid metabolism and DNA/RNA metabolism: proline, valine (957 cm^−1^), phenylalanine (1574 cm^−1^), amide-I (1654 cm^−1^), nucleic acid (1332 cm^−1^), and adenine (725 cm^−1^)	Blood serum samples from healthy volunteers (n = 22), patients with benign nodules (n = 19), and malignant nodules (n = 22)	PLS-DA
[77]	Thyroid nodules	Cytochrome *c* at 1600, 747, 1120–1128, and 1301 cm^−1^; carotenoids bands at 1003, 1155, and 1516 cm^−1^	Tissues of 30 subjects with thyroid nodules	AHCA and K-means analysis
[78]	Thyroid cancer	Lipid-containing components and populations of cytochrome-containing components	Human thyroid follicular carcinoma cells and human thyroid follicular epithelial cells	AHCA
[79]	Thyroid dysfunction	Ring vibration of glycerol at 630 cm^−1^ related to a decrease in lipid metabolism, C-C symmetric stretch of phenylalanine at 1004 cm^−1^, and vibrations of carotenoids at 1154 cm^−1^ and 1513 cm^−1^	Serum samples taken from 34 thyroid dysfunction patients and 40 healthy volunteers	PCA, SVM
[80]	Coronary heart disease	Peak at 1509 cm^−1^ that could be attributed to platelet-derived growth factor-BB	Urine samples from 87 patients with CHD and 20 healthy humans	PCA
[81]	Acute myocardial infarction	Phenylalanine (1000 cm^−1^) and tyrosine (825 cm^−1^)	Whole blood and blood serum samples from 10 patients and 10 healthy volunteers	PCA
[8]	Coronary heart disease	1223/1243/1272/1463/1481/1516/1536/1541/1550 cm^−1^	Urine samples of 157 patients and 63 healthy controls	
[82]	Myocardial viability	Heme proteins (755 cm^−1^) and collagen (the peak position about 2930 to 2950 cm^−1^)	Human hearts from five patients	PLS-DA
[83]	AD	I1342/I1243 ratio assigned to C-H deformation and amide III β-sheet, lactoferrin and lysozyme protein components	Tears from 18 AD patients, 8 patients with mild cognitive impairment, and 6 control volunteers	PCA
[84]	AD	Carotenoid level contribution at 1000, 1154, and 1519 cm^−1^	Blood serum sample from 26 healthy controls and 31 patients	PCA, PLS-DA, orthogonal PLS
[85]	AD	Nucleic acids, saccharides, and protein content	Serum samples from 10 patients with AD at either mild or 10 moderate stages, 5 patients with Lewy body dementia, 10 patients with PD dementia, 3 with frontotemporal dementia, and 10 healthy controls	ANN
[86]	PD	Proteins (829, 939, and 1001 cm^−1^ are signals from specific amino acids, while 1102 and 1346 cm^−1^ are due to the amide bands), nucleic acids (1244 cm^−1^), glycoproteins/saccharides (850 and 1444 cm^−1^), and lipids	Saliva of 23 PD patients, 10 AD patients, and 33 healthy controls	PCA, CNN
[87]	COVID-19	450 (thiocyanate), 1002 (phenylalanine), 1224 (β sheet structure in amide III), 1453 (lipid), 1586 (phenylalanine), and 2126 cm^−1^ (thiocyanate)	Saliva samples from nine healthy and COVID-19-infected subjects	Kernel PCA, SVM
[88]	COVID-19	Lipids, nitrogen compounds (urea and amines/amides) and nucleic acids, proteins, and amino acids (tryptophan)	Blood serum samples from 10 COVID-19-positive patients and 10 negative patients	PCA, PCA-DA, PLS-DA
[89]	COVID-19	1048 and 1126 cm^−1^ for an environment with a high content of aromatic amino acids, especially tryptophan and phenylalanine	30 currently SARS-CoV-2-infected patients (COV^ +^), 38 previously infected (COV^-^) patients, and 33 control subjects	PCA-LDA
[90]	COVID-19	proteins including albumin, lipids, phospholipids, and carotenoids, as well as nucleic acids, tryptophan, and immunoglobulins	54 control serum samples and 40 COVID-19 samples	PCA, PLS-DA
[91]	COVID-19	1317 cm^−1^ and 1432 cm^−1^ and between 2840 cm^−1^ and 2956 cm^−1^ that correspond to proteins and lipids	47 COVID infected patients	PCA, HCA, PLS

Note: PCA = principal component analysis; PLS analysis = partial least squares analysis/progression on the latent structures; LDA = linear discriminant analysis; DA = discriminant analysis; SVM = support vector machine; HCA = hierarchical component analysis; CNN = convolutional neural network, ANN = artificial neural network; AHCA = agglomerative hierarchical clustering analysis, AD = Alzheimer’s disease; PD = Parkinson’s disease.

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
