# Peer review of "Raman-Based Techniques in Medical Applications for Diagnostic Tasks: A Review"

_ijms, 2023, doi:10.3390/ijms242115605_

Round 1

Reviewer 1 Report

Manuscript ID: ijms-2672813 - Review Report

Raman-based Techniques in Diagnosing Socially Significant Diseases: a Review

By Yulia Khristoforova et al

The paper is acceptable with minor changes as following:

Page 2. Line 80. They should delete the and OH bonds in water, since OH-water is inert in Raman spectroscopy.

Page 9. The authors should rephrase the interpretation of Figure 3 because it does not make clear what they want to describe. The reader should read the reference to understand the fingure.

Author Response

We would like to acknowledge the reviewers for his work and provided critical comments. We made changes in the paper according to provided reviewers’ comments. All changes in the paper are highlighted with green color.

Below, please find point-by-point answers to reviewers’ comments.

Reviewer 1

1) Page 2. Line 80. They should delete the and OH bonds in water, since OH-water is inert in Raman spectroscopy.

Answer: The corresponding remark was corrected.

2) Page 9. The authors should rephrase the interpretation of Figure 3 because it does not make clear what they want to describe. The reader should read the reference to understand the fingure.

Answer: The corresponding correction was added to the 3.2. Section (355-360, 364-366 Lines).

Reviewer 2 Report

The authors report on the successful use of Raman based techniques in diagnosis or prognosis of critical conditions, such as cancer, viral infections and cardiovascular diseases. The combination with multivariate analysis and more modern data analysis approaches are discussed and analyzed as more feasible in clinical practice. The summarized studies are compelling and the cohort of data samples is in a relevant and reliable manner exploited. It is clear that the Raman/SERS-based methodology is a rapid, reproducible and valuable tool in minimum-invasive diagnosis with high accuracy of critical diseases. The schemes and tables are of good quality and add value to the work.

I would recommend that sections about ND are either mentioned as AD diagnosis, either completed with more data about other ND pathologies. The name of these sections is not reflecting the content totally. The authors either keep the AD data and name the sections as Alzheimer's either add more relevant content on this topic, including more other pathologies (see Biosensors 2023, 13, 499; https://doi.org/10.3390/bios13050499 and many other relevant reviews).

The title could also changed with one that actually reflects the reviewed work. It is not justified by the authors why these chosen diseases are most socially significant. So, please name differently these in the title and change it accordingly.

Author Response

We would like to acknowledge the reviewers for his work and provided critical comments. We made changes in the paper according to provided reviewers’ comments. All changes in the paper are highlighted with green color.

Below, please find point-by-point answers to reviewers’ comments.

1) I would recommend that sections about ND are either mentioned as AD diagnosis, either completed with more data about other ND pathologies. The name of these sections is not reflecting the content totally. The authors either keep the AD data and name the sections as Alzheimer's either add more relevant content on this topic, including more other pathologies (see Biosensors 2023, 13, 499; https://doi.org/10.3390/bios13050499 and many other relevant reviews).

Answer: We added review of 2 additional works studied the biomarkers of Parkinson’s disease and dopamine as biomarkers of serious neurological disorders (269-287 Lines).

2) The title could also changed with one that actually reflects the reviewed work. It is not justified by the authors why these chosen diseases are most socially significant. So, please name differently these in the title and change it accordingly.

Answer: The title was changed.